# Prevalence of prediabetes and type 2 diabetes mellitus in south and southeast Asian women with history of gestational diabetes mellitus: Systematic review and meta-analysis

**Chockalingam Shivashri**[1,2], **Wesley Hannah**[2], **Mohan Deepa**[2], **Yonas Ghebremichael-Weldeselassie**[1,3], **Ranjit Mohan Anjana**[2], **Ram Uma**[4], **Viswanathan Mohan**[2], **Ponnusamy Saravanan**[1,5]*

**1** Division of Populations, Evidence, and Technologies of Health Sciences, Warwick Medical School, University of Warwick, Coventry, United Kingdom, **2** Madras Diabetes Research Foundation and Dr. Mohan's Diabetes Specialities Centre, ICMR Center for Advanced Research on Diabetes, Chennai, Tamil Nadu, India, **3** School of Mathematics and Statistics, The Open University, Milton Keynes, United Kingdom, **4** Seethapathy Clinic & Hospital, Chennai, Tamil Nadu, India, **5** Department of Diabetes, Endocrinology, and Metabolism, George Eliot Hospital, Nuneaton, United Kingdom

* p.saravanan@warwick.ac.uk

## Abstract

### Background

The burden of Gestational Diabetes Mellitus (GDM) is very high in south Asia (SA) and southeast Asia (SEA). Thus, there is a need to understand the prevalence and risk factors for developing prediabetes and type 2 diabetes mellitus (T2DM) postpartum, in this high-risk population.

### Aim

To conduct a systematic review and meta-analysis to estimate the prevalence of prediabetes and T2DM among the women with history of GDM in SA and SEA.

### Methods

A comprehensive literature search was performed in the following databases: Medline, EMBASE, Web of Knowledge and CINHAL till December 2021. Studies that had reported greater than six weeks of postpartum follow-up were included. The pooled prevalence of diabetes and prediabetes were estimated by random effects meta-analysis model and $I^2$ statistic was used to assess heterogeneity.

### Results

Meta-analysis of 13 studies revealed that the prevalence of prediabetes and T2DM in post-GDM women were 25.9% (95%CI 18.94 to 33.51) and 29.9% (95%CI 17.02 to 44.57) respectively. Women with history of GDM from SA and SEA seem to have higher risk of developing T2DM than women without GDM (RR 13.2, 95%CI 9.52 to 18.29, p<0.001). The

**Data Availability Statement:** The dataset pertaining to our systematic review is available in

the Open Science Framework repository: osf.io/tfkh.

**Funding:** PS and YW are part funded by Medical Research Council, UK Grant number: MR/R020981/1 Funder: MRC- UK URL: https://mrc.ukri.org/funding/. SC's PhD scholarship is funded by the international doctoral training program (iDTP) by Novo Nordisk Plc, Copenhagen, Denmark based at The University of Warwick, Coventry, UK. The funders had no role in study design, data collection and analysis, decision to publish, or preparation of the manuscript.

**Competing interests:** The authors have declared that no competing interests exist.

subgroup analysis showed a rise in the prevalence of T2DM with increasing duration of follow-up.

## Conclusion

The conversion to T2DM and prediabetes is very high among women with history of GDM in SA and SEA. This highlights the need for follow-up of GDM women for early identification of dysglycemia and to plan interventions to prevent/delay the progression to T2DM.

## Introduction

There is a rapid increase in the prevalence of T2DM and the age of onset seems to be reducing globally [1]. It is estimated that 783 million adults will be affected by diabetes by 2045 [2]. South Asia (SA) and the southeast Asian (SEA) region are among the regions with the highest number of people having diabetes [3].

GDM is defined as glucose intolerance or hyperglycaemia that is first recognized or diagnosed in pregnancy [4]. It is estimated that GDM can affect more than 20 million live births every year [2, 4, 5]. Out of these, more than 90% are projected to occur in SA and SEA, which is one in every four live births [2, 5].

In addition to several short term maternal and offspring adverse outcomes, GDM contributes to adverse cardiometabolic outcomes for women in the long-term. These include T2DM, hypertension, and ischemic heart disease. The risk of developing T2DM among women with history of GDM can be up to 20-fold compared to healthy individuals [6, 7]. The prevalence of prediabetes post-GDM was observed to be between 3.9% and 50.9% based on the follow-up period after index delivery [8]. Similarly, the incidence of T2DM was reported to be between 2.6% and 70% from 6 weeks to 28 years postpartum [9], with the highest risk was observed around 3–6 years post-delivery [10]. Most of these studies were conducted in the western population and the studies in SA and SEA are limited, despite the higher prevalence of T2DM and GDM in these regions [3]. It has been suggested that ethnicity influences the rate of conversion to T2DM in women with GDM, but this claim has not been substantiated [11, 12]. Thus, this study aims to report on the prevalence and risk factors for prediabetes and T2DM in SA and SEA women with history of GDM by conducting a systematic review and meta-analysis.

## Methods

This review was conducted according to the Preferred Reporting Items for Systematic Reviews and Meta-Analyses (PRISMA) (S2 Appendix) [13]. The protocol for this systematic review and meta-analysis was registered with PROSPERO, and is available at https://www.crd.york.ac.uk/prospero/display_record.php?ID=CRD42020189654.

### Search strategy and selection criteria

A comprehensive literature search for observational studies that have followed-up women with history of GDM in SA and SEA, diagnosed by any criteria for GDM until end of December 2021 was performed in the following databases: Medline, EMBASE, Web of Knowledge and CINHAL. This was updated to include studies published until 30 June 2022. The search strategy included medical subject headings related to GDM (Gestational diabetes or Diabetes, Gestational, postpartum, Postpartum Period, postnatal or postnatal Care or post-natal) and T2DM (Diabetes Mellitus, Type 2/ or incidence of diabetes). A combination of these terms

was modified for specific bibliographic databases in combination with database-specific filters. The keywords/filters specific to each database for the included countries (SA–Afghanistan, Bangladesh, Bhutan, India, Maldives, Nepal, Pakistan, Sri Lanka; SEA–Brunei, Cambodia, Indonesia, Thailand, Malaysia, Philippines, Singapore, Vietnam) were used in the searches. A secondary search was also performed to identify studies that have reported on prediabetes as an individual outcome. The search was restricted to studies conducted in humans and published in English. The detailed search strategy is available in the S1 Appendix. An alert system was set up in these databases to identify any additional studies that got published between January 2022 and submission of this manuscript (30 June 2022).

Two independent reviewers (SC and HW) screened the titles and abstracts to identify appropriate studies. Full texts of articles from the relevant studies were reviewed and included according to the inclusion criteria: observational studies (prospective, retrospective, cross-sectional, case-control) that have reported on the postpartum follow-up of women with the history of GDM in SA and SEA. Conference proceedings and letters to the editor relevant to the inclusion criteria were also included. Studies that reported on randomized controlled trials conducted to either mitigate the risk of GDM or management of GDM and studies that had followed-up women for less than six weeks of postpartum were excluded. The reference lists of relevant studies were hand searched for additional eligible studies. Any disagreement between the reviewers was resolved by discussion with a third reviewer (DM). The details of the study selection process are presented in the PRISMA flowchart (Fig 1).

## Risk of bias and study quality assessment

The Newcastle-Ottawa scale (NOS) for cohort studies, proposed by Wells et al was used to assess the quality of the included studies [14]. This scale which is designed to appraise the quality of non-randomized studies by three categories: selection, comparability, and outcome. Each category has a set of numbered items to evaluate the study. For example, a maximum of one star can be awarded from the selection and outcome categories and two stars for the category of comparability. Overall, a study can be awarded from zero (low quality) to nine (high quality) stars. The risk of bias in cross-sectional studies was done by the using scale developed by Hoy and colleagues [15]. This included domains of sample selection, non-response bias, data collection, and measurement of reliability and validity. The risk of bias was reported as low, medium, and high-risk for each category. Publication bias was evaluated using Egger's test [16].

## Data synthesis and statistical analysis

Data extraction was done by the two independent reviewers (SC and HW), by extracting the study characteristics that included information on study design, country of study, diagnosis of GDM and T2DM, characteristics of the women (age, BMI) at the baseline and at the follow-up period. When two studies reported outcomes from the same cohort, the study with more complete information related to this review was included for the analysis.

Sub-group analyses were carried out based on the diagnostic criteria of GDM, duration of follow-up and type of studies. As the International Association of Diabetes and Pregnancy Study Groups (IADPSG) criteria [17] is increasingly used across the world, we assessed the prevalence of prediabetes and T2DM by IADPSG vs. other criteria. Duration of follow-up is split into less than two, between two to five and more than five years of duration for the prevalence of T2DM and less than two and more than two years for prediabetes. Type of studies were split into prospective vs. other type of studies.

The Metafor package was used for quantitative synthesis [18]. The inverse variance method was used to estimate the pooled prevalence expressed as the proportion of women with history

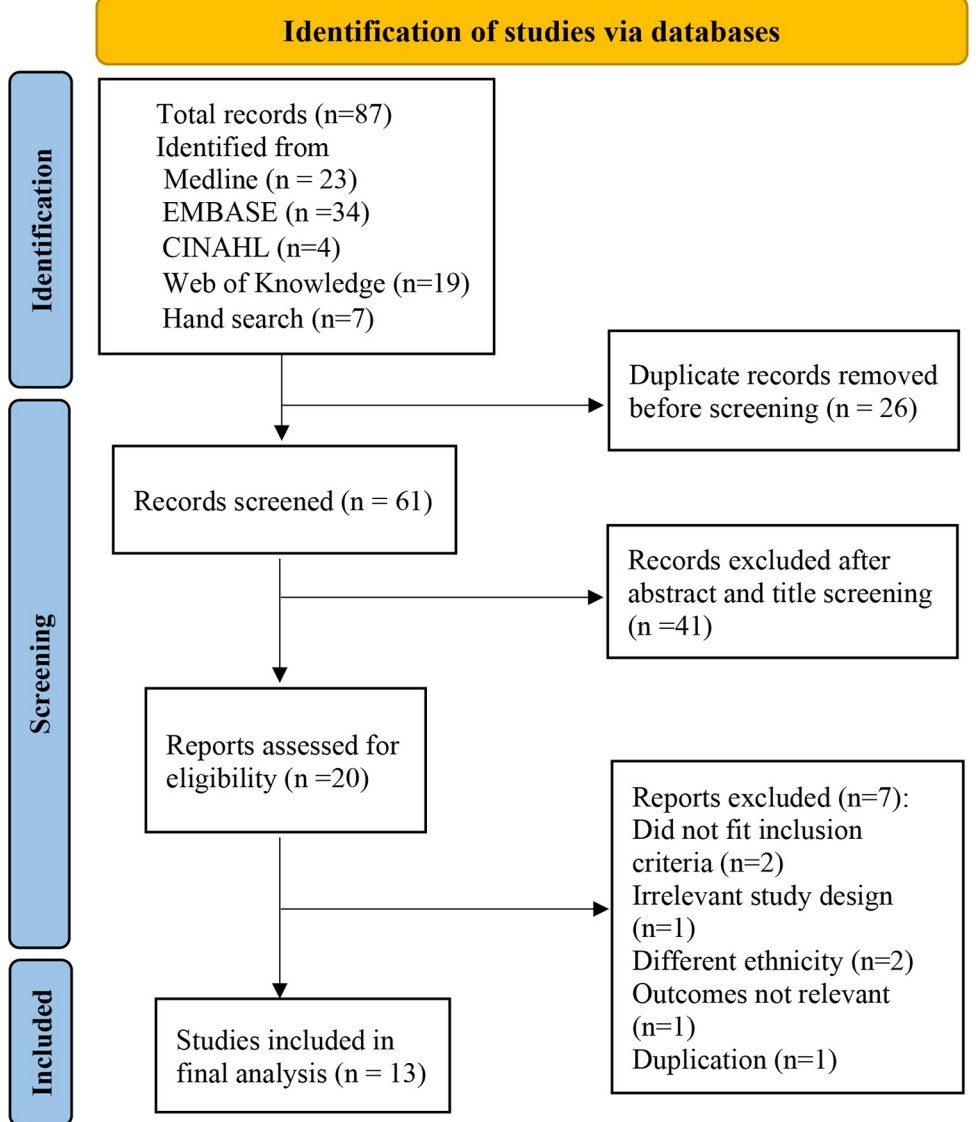

**Fig 1. PRISMA flow diagram showing the study selection process.**

of GDM who have developed T2DM/prediabetes. The study results were pooled by using random effects analysis models fitted in 'R' programming language [19]. The heterogeneity observed between the studies was estimated using the $I^2$ statistic [16] which describes the percentage of variation not due to sampling errors between the studies. Funnel plots were developed to assess the publication bias among the studies. The asymmetry of the funnel plots was investigated using the Egger's test. A sensitivity analysis was performed to investigate the effect of each study on the pooled prevalence. All analyses were done in 'R' software [19].

## Results

### Identification of studies

The electronic database search and the hand search together yielded 87 studies of which 26 studies were removed after evaluation for duplicate records. Sixty-one studies were then

selected, of which 41 were excluded after initial title and abstract screening. Full text screening excluded seven studies because of the following reasons: different ethnicity, article on protocol, reported only about risk factors, an intervention study, conference proceedings with no baseline data, reported only on the uptake rates of postnatal OGTT screening and a conference proceeding which was a duplication of a subsequent full manuscript with more relevant data. Thus, thirteen studies were included for the final analyses (Fig 1).

## Study characteristics

Among the 13 studies, seven were from India, two from Singapore and one each from Sri Lanka, Pakistan, Malaysia, and Thailand. The studies showed varied lengths of follow-up and diagnostic criteria for GDM. The diagnosis of T2DM was by WHO criteria. Prediabetes was defined as impaired fasting glucose (IFG) or impaired glucose tolerance (IGT) or combined IGT and IFG. The diagnosis of T2DM was predominantly by using oral glucose tolerance test except for one study which used HbA1c [38]. The follow-up period ranged between 0.25 to 15 years. Summary of study characteristics that reported the prevalence of prediabetes and/or diabetes are provided in Table 1. Summary of key maternal risk factors are shown in Table 2.

**Table 1. Characteristics of studies that reported the prevalence of prediabetes and T2DM.**

| Study | Study design | Country of study | Criteria of GDM diagnosis | Total GDM nos. (n) | GDM to T2DM (n) | GDM to Prediabetes (n) | Non-GDM nos. (n) | Non-GDM to T2DM (n) | Prevalence of T2DM (%) | Prevalence of prediabetes (%) | Follow-up (years) | Response rate (%) |
|---|---|---|---|---|---|---|---|---|---|---|---|---|
| Dai et al 2022 [33] | Retrospective | Singapore | IADPSG | 942 | 33 | 124 | NR | NR | 3.5 | 13.2 | 0.25 | 45.1 |
| Hewage et al 2021 [34] | Prospective | Singapore | WHO 1999 | 117 | 13 | 38 | NR | NR | 11.2 | 32.8 | 5.0 | NR |
| Aziz et al 2018 [35] | Prospective | Pakistan | IADPSG | 27 | 11 | NR | NR | NR | 41.0 | NR | 2.0 | 35.8 |
| Goyal et al 2018 [36] | Cross sectional | India | IADPSG | 267 | 28 | 126 | NR | NR | 10.5 | 47.2 | 1.67 | 31.4 |
| Herath et al 2017 [37] | Retrospective | Sri Lanka | WHO 1999 | 119 | 73 | NR | 456 | 18 | 61.0 | NR | 10.9 | >70 |
| Gupta et al 2017 [38] | Prospective | India | CC and IADPSG | 366 | 119 | 144 | NR | NR | 33.0 | 39.3 | 1.16 | 37 |
| Bhavadharani et al 2016 [26] | Prospective | India | IADPSG | 203 | 7 | 34 | NR | NR | 3.0 | 16.8 | 1.0 | 95.8 |
| Jindal et al 2015 [39] | Prospective | India | IADPSG | 62 | 4 | 17 | NR | NR | 6.0 | 27.4 | 0.25 | 82.7 |
| Mahalakshmi et al 2014 [40] | Retrospective | India | CC | 174 | 101 | 19 | NR | NR | 58.0 | 10.9 | 4.5 | NR |
| Youngwanichsetha and Phumdoung 2013 [41] | Cross sectional | Thailand | ADA 2010 | 210 | NR | 56 | NR | NR | NR | 27.6 | 0.12 | NR |
| Chew et al 2012 [42] | Cross sectional | Malaysia | WHO 1984 | 342 | 159 | 117 | NR | NR | 46.5 | 34.2 | 0.25–15 | NR |
| Krishnaveni et al 2007 [43] | Prospective | India | CC | 35 | 13 | 11 | 489 | 8 | 37 | 31.4 | 5 | 66.7 |
| Kale et al 2004 [44] | Prospective | India | WHO 1985 | 125 | 65 | 19 | 240 | 14 | 52 | 15.2 | 4.5 | 69.2 |

ADA—American Diabetes Association

CC-Carpenter & Coustan

IADPSG—International Association of Diabetes and Pregnancy Study Group

WHO—World Health Organisation

NR- Not reported

## Assessment of study quality

The quality appraisal revealed that out of the 10 cohort studies, three had a low risk of bias, four showed unclear risk and the remaining three showed a high risk of bias. All three cross-sectional studies showed a low risk of bias (S1 Fig). The detailed quality assessment of studies is provided in S1 and S2 Tables.

**Table 2. Maternal characteristics reported by the studies included in the systematic review.**

| Study | Study design | Country | Maternal age at follow-up (yr) | BMI or weight at index pregnancy | BMI or weight at follow-up |
|---|---|---|---|---|---|
| Dai et al 2022 [33] | Retrospective | Singapore | 32.7±4.7 | <18.5 (n = 15) 18.5–24.9 (n = 245) 25–29.9 (n = 235) ≥30 (n = 206) | NR |
| Hewage et al 2021 [34] | Prospective | Singapore | Normoglycemia 32.8±4.5## Dysglycemia$ 33.9±5.2 | Normoglycemia 22.7 ±3.8 Dysglycemia$ 25.0 ±4.3 | Normoglycemia <23–27, 23 to <27.5–20, ≥ 27.5–5 Dysglycemia <23–13, 23 to <27.5–14, ≥ 27.5–15 |
| Aziz et al 2018 [35] | Prospective | Pakistan | 28.94±2.84 | **GDM** 69.5± 8.22 kg **Non-GDM** 56.54 ±5.42 kg | **GDM** 73.26±6.86 kg **Non-GDM** 67.23±4.65 kg |
| Goyal et al 2018 [36] | Cross-sectional | India | Normoglycemia 31.3±4.4 Dysglycemia^ 33.3±4.5 | NR | Normoglycemia <25–44, 25–29.9–50, ≥30–19 Dysglycemia^ <25–47, 25–29.9–54, ≥30–53 |
| Herath et al 2017 [37] | Retrospective | Sri Lanka | **GDM** 42.7±5.37 **Non-GDM** 38.7±5.36 | **GDM** < 18.5–1 18.5–24.9–39 >25–28–28 | NR |
| Gupta et al 2017 [38] | Prospective | India | 30.2±4.9 | 23.6±4.7* | 27.6±5.2 |
| Bhavadharani et al 2016 [26] | Prospective | India | Dysglycemia^ 29.6±4.2 Normoglycemia 28.6±4.3 | Dysglycemia^ 28.0±5.0 Normoglycemia 25.8±4.7 | NR |
| Jindal et al 2015 [39] | Prospective | India | Normoglycemia 32.24±3.60, Dysglycemia^ 31±3.50 | NR | NR |
| Youngwanichsetha and Phumdoung 2013 [41] | Cross sectional | Thailand | 34.54 | NR | 25.0–29.9–34 30.0–39.9–22 |
| Mahalakshmi et al 2014 [40] | Retrospective | India | 29±4 | 28.6±4.1 | NR |
| Chew et al 2012 [42] | Cross sectional | Malaysia | Normoglycemia 37.6±5.3, IGT 37.7±5.0, IFG 38.9±5.6, Combined IFG/IGT 39.7 ±6.8, T2DM 39.4±4.5 | NR | Normoglycemia 25.69±4.85, IGT 26.59 ±4.84, IFG 26.22±4.33, Combined IFG/IGT 28.53±5.07, T2DM 30.26±4.62 |
| Krishnaveni et al 2007 [43] | Prospective | India | **GDM:** Normoglycemia-32.2, IFG/IGT-34, T2DM-33.5, **Non-GDM:** Normoglycemia -28.1, IFG/IGT-29.3, T2DM- 28.6 | NR | **GDM:** Normoglycemia -23.6, IFG/IGT -26.1, T2DM—26.7, **Non-GDM:** Normoglycemia-23.2, IFG/IGT-24.8, T2DM—28.9 |
| Kale et al 2004 [44] | Prospective | India | **GDM:** Normoglycemia- 33, IGT– 33, T2DM—34 | NR | **GDM:** Normoglycemia -25.8, IGT—25.4, T2DM- 26.2 |

IFG- Impaired fasting glucose

IGT–Impaired glucose tolerance

^—includes isolated IFG + isolated IGT + combined IFG/IGT + diabetes

$—includes diabetes and prediabetes

*—prepregnancy BMI, ##- age at delivery

NR- Not reported

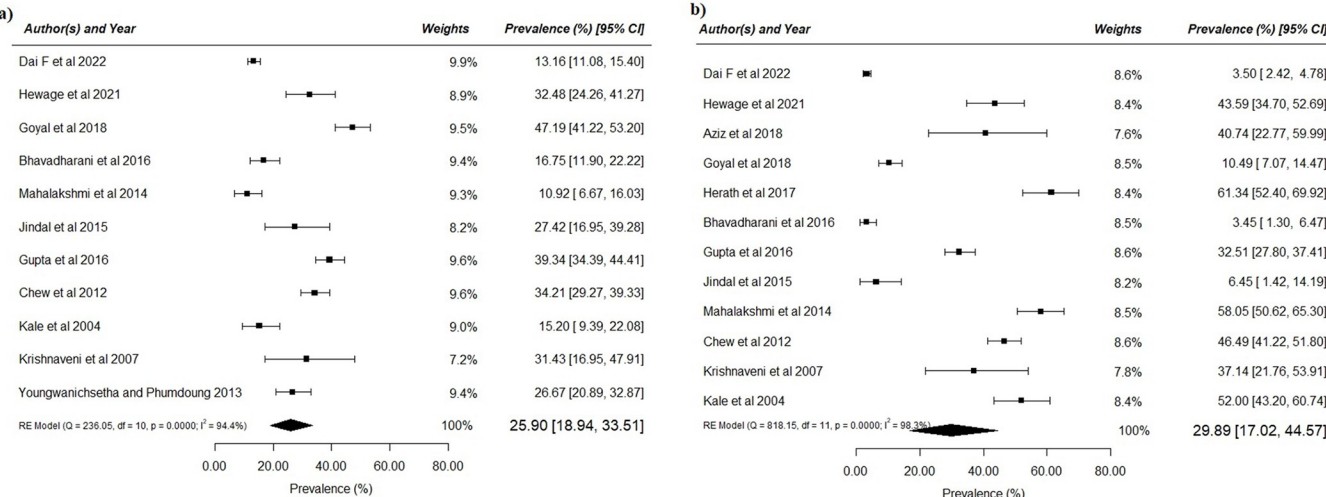

**Fig 2.** Prevalence of (a) prediabetes and (b)T2DM in women with history of GDM in SA and SEA.

## Prevalence of prediabetes

Eleven studies have reported the rates of prediabetes (IFG or IGT and combined IGT and IFG) among the participants with previous GDM. This included a total of 2843 participants. The pooled prevalence of prediabetes was 25.9% (95% CI 18.94, 33.51) (Fig 2A). Significant heterogeneity among studies was observed ($I^2 = 94.3\%$, p<0.001). Of these 11 studies, seven were from India (Table 1).

## Prevalence of T2DM

Twelve studies reported the prevalence of T2DM. This included a total of 2779 participants. The pooled prevalence of T2DM was 29.9% (95% CI, 17.02, 44.57) (Fig 2B). Significant heterogeneity was observed ($I^2 = 98.3\%$, p<0.001).

## Sensitivity analysis

Because of the high heterogeneity, we carried out a sensitivity analysis to estimate the pooled prevalence of prediabetes and T2DM, by excluding one study at a time. For prediabetes, this ranged between 25.0% and 27.7%. For T2DM, this was 27.2% to 33.4%. These estimates were close to the overall prevalence estimates of 25.9% and 29.9% for prediabetes and T2DM, respectively, when all the studies were included. This indicates that no single study had significantly influenced the overall estimates (S2 Fig).

## Subgroup analyses

Subgroup analyses were performed to assess whether diagnostic criteria of GDM or the duration of follow-up have any differential effect on the prevalence of T2DM/prediabetes

**Based on follow-up period.** For prediabetes, 12 studies have reported the follow-up period. These were grouped as less (seven studies, n = 2220) or more (five studies, n = 623) than two years of follow-up. No significant difference was seen in the pooled prevalence of prediabetes (less than 2 years: 27.2%; 95%CI: 18.97, 36.27; more than 2 years: 23.9%, 95%CI: 14.14, 35.26; p = 0.65) (Fig 3A).

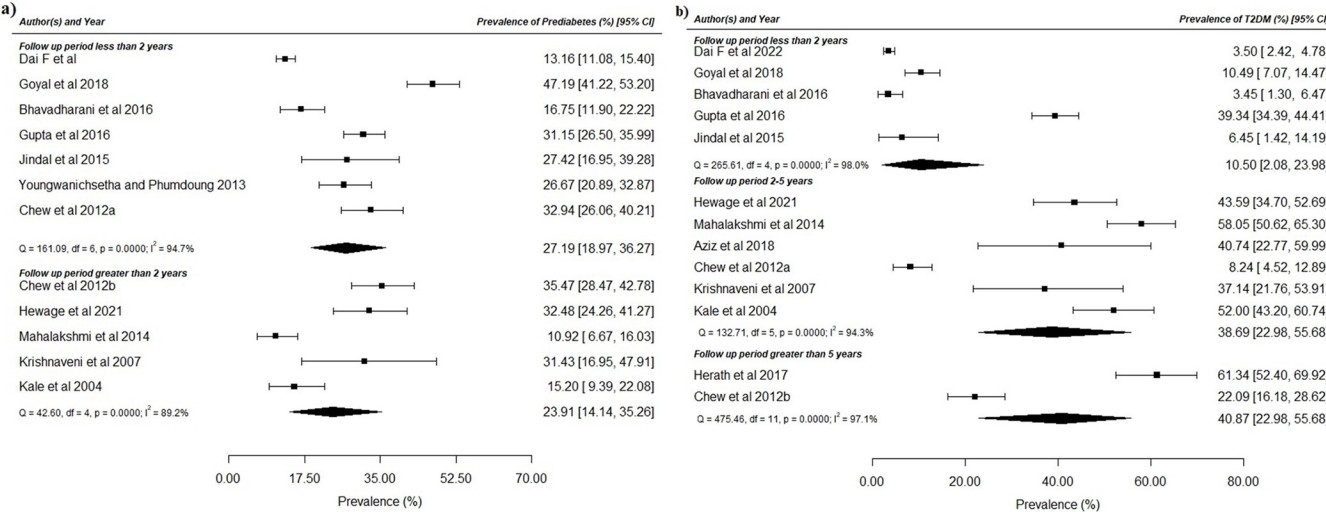

**Fig 3.** Prevalence of (a) prediabetes and (b) T2DM based on the follow-up period.

For T2DM, these were categorized into less than two years (five studies, n = 1840), 2–5 years, (six studies, n = 648) and greater than five years (two studies, n = 291) of follow-up period. Compared to the less than 2 years follow-up period (10.5%; 95%CI: 2.08, 23.98), the prevalence of T2DM was significantly higher in the 2–5 years group (38.7%; 95%CI: 22.98, 55.68; p<0.0001) and more than 5 years group (40.9%; 95%CI: 22.98, 55.68; p = 0.003) (Fig 3B).

**Based on diagnostic criteria of GDM.** The studies were categorized based on IADPSG vs other criteria. The prevalence of prediabetes by IADPSG criteria (five studies, n = 1542) was 27.1% (95% CI, 15.25, 40.85) compared to 26.5% (95% CI, 18.45, 35.45) by other criteria (seven studies, n = 1301) (p = 0.28) (S3 Fig). The prevalence of T2DM by IADPSG criteria (six studies, n = 1569) was 14.6% (95% CI, 3.31, 31.51) compared to 47.9% (95% CI, 40.47, 55.45) by other criteria (seven studies, n = 1210) (p = 0.004) (S3 Fig).

**Based on study design.** Among the studies included in the review, seven studies had a prospective study design. The prevalence of prediabetes and T2DM for prospective studies were 24.0% (95% CI: 17.09, 31.96) and 22.0% (95%CI: 0.00, 65.44) respectively. The prevalences of other study design were 23.81% (95% CI: 12.67, 37.13) and 30.27% (95% CI: 8.91, 57.55) for prediabetes and T2DM, respectively. There was no significant difference observed based on the study design for both prediabetes (p = 0.985) or T2DM (p = 0.590) (S4 Fig).

## Relative risk of T2DM

Overall, only three studies (n = 1465) have reported the prevalence of T2DM in GDM and non-GDM mothers (Table 1) Relative risk of developing T2DM were calculated based on the reported prevalence (S3 Table). Women with history of GDM were at 13 times higher risk of developing T2DM than women without the history of GDM (RR13.2, 95%CI 9.52 to 18.29, p<0.001). There was no heterogeneity observed among these studies ($I^2$ = 0.0%, p = 0.2969).

## Risk factors for the development of prediabetes and diabetes

Unfortunately, not many studies reported the risk factors (such as age and BMI and family history) for the development of prediabetes and diabetes. While most reported age and reported BMI at the time of follow-up, none reported their impact on the onset of prediabetes/diabetes post GDM (Table 2).

### Assessment of publication bias

There was no indication of publication bias, using Egger's tests (p = 0.38). The funnel plot of 13 studies included in the meta-analysis is provided in S5 Fig.

## Discussion

This systematic review and meta-analysis involving women with the previous history of GDM living in SA and SEA revealed a pooled prevalence of 29.9% for T2DM (n = 2779) and 25.9% for prediabetes (n = 2843) at postpartum follow-up. SA and SEA women with history of GDM had 13-times higher risk of developing T2DM compared to those without the history of GDM.

To the best of our knowledge, this is the first study to quantify the risk of prediabetes and T2DM among women with GDM in SA and SEA. Earlier systematic reviews published in 2009 and 2020 have reported 7-times and 10-times higher risk of conversion to T2DM among women with history of GDM involving all ethnic groups [6, 7]. These studies reported no difference across ethnicities as they were underpowered to observe any ethnic differences. Our findings reveal that SA and SEA women with GDM are at higher risk of conversion to T2DM compared to global estimates. Our findings reveal that SA and SEA women with GDM are at higher risk of conversion to T2DM compared to global estimates. However, this is based on limited number of studies. Among 13 studies included in this systematic review, only three had high risk of bias. These findings need to be confirmed through well-designed longitudinal studies that also controls for other risk factors (such as age, BMI, and family history) to assess the independent role of GDM in these ethnic groups.

The SA and SEA ethnicities have higher predisposition to T2DM compared to other ethnic groups [20]. Anjana et al [21] has reported that the age of onset of T2DM among South Asians is earlier compared to other ethnic groups. In addition to the pooled prevalence rate of 29.9% for T2DM, which is higher beyond the first 2 years of postpartum, rate of prediabetes is also high (25.9%) in women with history of GDM. These rates are much higher than previously reported and in other populations [22, 23]. Thus, GDM might be a key contributing factor for the decreasing age of onset of T2DM in these population, at least for women.

The combined prevalence of more than 50% of prediabetes and T2DM highlights the importance of improving the postpartum screening for all forms of dysglycemia in women with recent history of GDM. However, the uptake rate of postpartum glucose monitoring is sub-optimal even in well developed countries [24]. In addition, there seem to be limited evidence to compare the prevalence of T2DM post-GDM in different ethnicities including PIMA Indians. A recent study by Napoli et al [25] in Italy, reported that only 34.4% of women from 'STRONG' observational study underwent postpartum glucose monitoring. We also observed that only three studies had a follow-up rate of more than 70%, although we have showed earlier that it is feasible at least in research settings, with a follow up of 95.8% [26]. Whether this can be replicated in real-world settings in SA and SEA will require additional studies. A targeted approach for postpartum screening may be a better approach similar to a study by Nishanthi et al. [27] A simple machine learning approach using the routinely available antenatal factors could may identify who is unlikely to attend postpartum screening and enable better, targeted follow-up. The simple risk calculator proposed by Nishanthi et al [27] is easy to use for healthcare providers. However, the validity of this in SA and SEA population is not proven. A similar strategy for predicting the onset of prediabetes and T2DM can offer prevention strategies to be implemented in post-GDM women. In addition, with high birth rates in these countries [28], such interventions in between pregnancies can also reduce the risk of recurrence of GDM. Such 'inter-pregnancy' preventive interventions could be vital in reducing adverse metabolic

programming in the offspring [5]. With the effective strategies (both lifestyle and metformin) available in these populations [29], such interpregnancy interventions are urgently warranted.

The subgroup analyses revealed that different diagnostic criteria (lower rates of T2DM but higher prediabetes rates if IADPSG criteria was used to diagnose GDM) influence the rates of dysglycemia postpartum. It is conceivable that IADPSG detects a 'milder' form of dysglycemia in pregnancy than other criteria, albeit important for the short-term adverse outcomes in pregnancy. The stratification based on study design showed no significant difference in the prevalence of prediabetes and T2DM. However, only three cohort studies have prospectively followed up women with history of GDM and the non-GDM group.

## Strengths and limitations

This is the first study to report on the population- specific prevalence of T2DM and prediabetes in women with GDM in SA and SEA, where the metabolic burden in young adults is very high. Our study has important limitations. First, while there were adequate number of women for estimating the prevalence rates, the relative risk estimation is based on only three studies and there were not enough studies to assess the prevalence for each country within these regions. Second, as risk factors contributing to the prevalence of T2DM are not reported in most studies, we are not able to assess the contribution of individual risk factors to the development of T2DM, post-GDM. Third, none of the studies reported the influence of rapid urbanization, which is a major contributing factor to differential rates of T2DM in these countries. Finally, we did not find studies that reported other co-existing cardiometabolic disorders such as hypertension, dyslipidemia and cardiovascular disorders.

## Implications for research and clinical practice

Our study highlights the gaps in the existing evidence in SA and SEA countries where the prevalence of both GDM and T2DM are high. The high rates of prediabetes and diabetes in these populations soon after an index pregnancy with GDM, suggest the importance of postpartum testing and early detection of T2DM. Despite the existence of guidelines [30–32] over a long period, only limited women with history of GDM adhere to postpartum glucose screening, due to several barriers [24]. Our findings raise the following key questions for future research: 1) What are the barriers/enablers for glucose testing post-GDM in different SA and SEA countries? 2) What are the risk factors (including modifiable risk factors such as postpartum weight retention) in women with history of GDM that contribute to the high prevalence of prediabetes and T2DM? 3) Do other cardiometabolic disorders co-exist in these women, similar to studies reported in western populations? 4) Can we develop an individualised prediction of incident T2DM post-GDM? and 5) Can we develop personalised strategies for interpregnancy interventions for prevention of GDM and subsequent T2DM?

## Conclusion

Women living in SA and SEA countries with GDM have high rates of prediabetes and T2DM on postpartum follow-up. Despite the lack of adequate data, which requires carefully designed longitudinal studies, these findings highlight the importance of prioritising women with history of GDM for T2DM prevention strategies. Development of precision medicine would be key for individualised strategies, which is likely to have better adherence rates for both screening and prevention.

### Patient and public involvement

Patients and/or the public were not involved in the design or conduct or reporting or dissemination plans of this research.

## Supporting information

**S1 Fig. Study quality appraisal of the included studies.**
(TIF)

**S2 Fig.** Sensitivity analyses for the prevalence of (a) prediabetes and (b) T2DM.
(TIF)

**S3 Fig.** Prevalence of (a) prediabetes and (b) T2DM based on the diagnostic criteria of GDM.
(TIF)

**S4 Fig.** Prevalence of (a) design prediabetes and (b) T2DM based on the study.
(TIF)

**S5 Fig. Funnel plot for publication bias.**
(TIF)

**S1 Table. Quality assessment of cohort studies using Newcastle Ottawa scale (n = 10).**
(DOCX)

**S2 Table. Quality assessment of cross-sectional studies (n = 3).**
(DOCX)

**S3 Table. Relative risk of T2DM in women with history of GDM compared with healthy controls in SA and SEA.**
(DOCX)

**S1 Appendix. Detailed search strategy.**
(DOCX)

**S2 Appendix. PRISMA checklist.**
(DOCX)

## Author Contributions

**Conceptualization:** Chockalingam Shivashri, Ponnusamy Saravanan.

**Data curation:** Chockalingam Shivashri, Wesley Hannah, Mohan Deepa.

**Formal analysis:** Chockalingam Shivashri, Wesley Hannah, Mohan Deepa, Yonas Ghebremichael-Weldeselassie, Ranjit Mohan Anjana, Ram Uma, Viswanathan Mohan, Ponnusamy Saravanan.

**Funding acquisition:** Ponnusamy Saravanan.

**Methodology:** Chockalingam Shivashri, Ponnusamy Saravanan.

**Resources:** Ponnusamy Saravanan.

**Software:** Chockalingam Shivashri, Yonas Ghebremichael-Weldeselassie.

**Supervision:** Ponnusamy Saravanan.

**Validation:** Yonas Ghebremichael-Weldeselassie, Ponnusamy Saravanan.

**Visualization:** Chockalingam Shivashri, Yonas Ghebremichael-Weldeselassie.

**Writing – original draft:** Chockalingam Shivashri.

**Writing – review & editing:** Chockalingam Shivashri, Wesley Hannah, Mohan Deepa, Ranjit Mohan Anjana, Ram Uma, Viswanathan Mohan, Ponnusamy Saravanan.

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
