## [Decision Letter · Decision Letter 0]

2 Oct 2022

PONE-D-22-18537Prevalence of prediabetes and type 2 diabetes mellitus in south and southeast Asian women with history of gestational diabetes mellitus: systematic review and meta-analysis

PLOS ONE

Dear Prof. Saravanan,

Thank you for submitting your manuscript to PLOS ONE. After careful consideration, we feel that it has merit but does not fully meet PLOS ONE’s publication criteria as it currently stands. Therefore, we invite you to submit a revised version of the manuscript that addresses the points raised during the review process.

We look forward to receiving your revised manuscript.

Kind regards,

Diane Farrar

Academic Editor

PLOS ONE

Journal Requirements:

“PS and YW are part funded by Medical Research Council, UK Grant number: MR/R020981/1 Funder: MRC- UK URL: https://mrc.ukri.org/funding/.This systematic review was funded by the international doctoral training program (iDTP) by the University of Warwick, Coventry, UK. The iDTP was funded by an unrestricted PhD scholarship scheme for Indian graduates by Novo Nordisk Plc, Copenhagen, Denmark.”

6. We note that this manuscript is a systematic review or meta-analysis; our author guidelines therefore require that you use PRISMA guidance to help improve reporting quality of this type of study. Please upload copies of the completed PRISMA checklist as Supporting Information with a file name “PRISMA checklist”.

Reviewers' comments:

Reviewer's Responses to Questions

**Comments to the Author**

1. Is the manuscript technically sound, and do the data support the conclusions?

Reviewer #1: No

Reviewer #2: Yes

2. Has the statistical analysis been performed appropriately and rigorously? 

Reviewer #1: No

Reviewer #2: Yes

3. Have the authors made all data underlying the findings in their manuscript fully available?

Reviewer #1: Yes

Reviewer #2: Yes

4. Is the manuscript presented in an intelligible fashion and written in standard English?

Reviewer #1: No

Reviewer #2: Yes

5. Review Comments to the Author

Reviewer #1: The authors report the prevalence of pre-diabetes and diabetes among women with prior GDM. The main challenge in understanding these data is the high level of heterogeneity. The heterogeneity is not surprising given the variation in populations, in follow-up time and in diagnostic criteria of both GDM and of the outcomes of T2DM and pre-diabetes. This heterogeneity precludes simple summary statistics such as overall prevalence. It makes no sense to average the prevalence of diabetes in cohorts that are so different (eg 27 women followed for two years in Pakistan with 342 women followed for up to 15 years in Malaysia), since the average can’t be applied to anyone. A narrative description with some attempt to group by important risk factors is the approach needed here. This also relates to the study aims, which indicate an intention to estimate pooled prevalence, when such an outcome was never likely to be possible. Even if the attempt was to estimate a prevalence for the region, it can only have meaning when stratified by follow-up time, and depends on getting studies from each country in the region. Furthermore, weighting the prevalence estimates by sample size (as per meta-analysis) does not help in estimating the prevalence in the region, as a large study from a small country will skew results towards the small country.

I couldn’t find the supplementary tables.

I was concerned about the quality and representativeness of some of the studies. Three studies followed fewer than 100 women with GDM, and it seems rather unlikely that such studies have an adequate epidemiological design to provide T2DM incidence in a manner that can be generalized. For example, reference 35 provides data on only 27 women. Looking at the paper itself, it becomes apparent that the recruitment was from a tertiary hospital, and that of the 78 women identified with GDM, only 27 were re-screened for T2DM. The 78 is likely a biased sample of the general GDM population, and the 27 are likely a biased sample of the 78. Such a study doesn’t contribute usefully to the authors’ aims. How carefully have other studies been reviewed?

Abstract results. ‘The relative risk of T2DM among women with history of GDM from SA and SEA was 13 times higher than’. A relative risk can’t be higher in one group than in another. Only the risk can be higher.

Line 68-69. ‘The rate of prediabetes among these women was observed to be between 3.9% and 50.9%’. The word ‘rate’ implies a time component, but these look like prevalence data. Please be more precise with language.

Line 71. ‘during 6 weeks postpartum to 28 years postpartum’. Replace ‘during’ with ‘from’.

Line 229. Relative risk of diabetes appears to have been calculated from the published data, and presumably therefore was unable to adjust for confounders such as age. If this is the case, then the RR is quite possibly very misleading, and should not be presented.

Line 249. ‘indicating that it lies closer to the pooled overall prevalence’. Closer than what?

English language use needs to be improved throughout by a native English speaker.

Reviewer #2: The growing prevalence of type 2 diabetes in SA and SEA regions is a major concern. Gestational Diabetes is well recognised as a major risk factor for future diabetes. However, the lack of data in SA /SEA populations until recently, has been a major limitation to understand the true impact of this problem and to develop effective interventions. Improved understanding of the risks and opportunities to mitigate this can have he benefits for the prevention of T2DM and maternal health. This systematic review and meta- analysis addresses this important issue and provides a fairly good insights on the risk of T2DM in patients with GDM. The work presented here is of good quality but there are a few points that need clarification.

1. Abstract : This is good overall. It would be good to provide some data/figures in relation to the last sentence of the results section.

2. Introduction: It is well written and references and places the research in context.

3. Methods: A lot of detail is provided and the essential ,principles of systematic review have been followed. PRISMA guidelines have been adhered to. I would suggest it would be better to describe the combinations of SEARCH terms used to identify the relevant papers.

It is surprising to see that Chinese data was not considered in this study. Why? What group would Chinese be categorised under by the authors?

Although SA and SEA share many common characteristics, many would regard them as distinct ethnic groups. What is the rationale in combining them?

As evident from the data and also the meta analysis, there is considerable heterogeneity between the studies. Moreover , including retrospective, cross sectional and prospective studies together actually weakens the study. Wonder if it would be more prudent to focus on prospective studies that have a nonGDM comparison group.

It is common experience that most GDM patients do not attend follow up. It is therefore difficult to ascertain if all those with GDM were assessed for the duration of the studies and if they had a test to confirm the glycemic status. Lot of attrition can be expected in these situations and can therefore lead to under or over estimation of the true risk. No data is provided regarding this. Also, were there any other tests eg HbA1c used to diagnose pre-diabetes and diabetes?

Given the high prevalence of T2DM in these populations, it is possible that some of the subjects classed as GDM may in fact be T2DM or PRE DIABETES prior to pregnancy. How was this addressed in the studies?

For the 3 studies that were used for computing Relative Risk, was that based on incident Pre-diabetes or Diabetes?

It would be good to see some data on additional risk factors if available eg parity, family history?

A risk score /calculator I mentioned in the manuscript but I could not find it ( apologies if I missed it).

Discussion

The discussion is well written and balanced. It can be improved by some reference to data in other populations and how these findings compare by providing some figures and data. More so with reference to other ethnic groups with similarly high pre disposition to T2DM such as PIMA Indians etc.

6. PLOS authors have the option to publish the peer review history of their article (what does this mean?). If published, this will include your full peer review and any attached files.

Reviewer #1: No

Reviewer #2: **Yes: **Srikanth Bellary

---

## [Author Response · Author response to Decision Letter 0]

16 Nov 2022

Response to Reviewers

Reviewer #1:

Comments: The authors report the prevalence of pre-diabetes and diabetes among women with prior GDM. The main challenge in understanding these data is the high level of heterogeneity. The heterogeneity is not surprising given the variation in populations, in follow-up time and in diagnostic criteria of both GDM and of the outcomes of T2DM and pre-diabetes. This heterogeneity precludes simple summary statistics such as overall prevalence. It makes no sense to average the prevalence of diabetes in cohorts that are so different (eg 27 women followed for two years in Pakistan with 342 women followed for up to 15 years in Malaysia), since the average can’t be applied to anyone. A narrative description with some attempt to group by important risk factors is the approach needed here. This also relates to the study aims, which indicate an intention to estimate pooled prevalence, when such an outcome was never likely to be possible. Even if the attempt was to estimate a prevalence for the region, it can only have meaning, when stratified by follow-up time, and depends on getting studies from each country in the region. Furthermore, weighting the prevalence estimates by sample size (as per meta-analysis) does not help in estimating the prevalence in the region, as a large study from a small country will skew results towards the small country. 

Response: We thank the reviewer for this comment. Yes, we completely agree that the studies included in this systematic review have high level of heterogeneity. To address the heterogeneity, we tried to include sensitivity analysis by removing one study at a time (Supplementary Figure 5), stratifying the studies based on diagnostic criteria (Supplementary Figure 2) and follow-up period (Supplementary Figure 3). However, it is unfortunate that these supplementary Tables/Figures were not accessible to the reviewer. We have uploaded all files and our system shows all supplementary materials. Apologies for the inconvenience caused and we have now highlighted this issue to the Editor.

Taking the reviewer’s comment on board, the sub-group analysis on follow-up period is now moved to the current Figure 3, in the revised version of the manuscript. Further, the results of the sensitivity analysis, shows that the observed prevalences (25% to 27.7%;27.2% to 33.4%) are close to the pooled prevalences of prediabetes (25.9%) and T2DM (29.9%), respectively. We hope that this partly addresses the issue of heterogeneity. We have made this explicit and reflected this in the results section in our revised manuscript (line 199-203).

Comments: I couldn’t find the supplementary tables. I was concerned about the quality and representativeness of some of the studies. Three studies followed fewer than 100 women with GDM, and it seems rather unlikely that such studies have an adequate epidemiological design to provide T2DM incidence in a manner that can be generalized. For example, reference 35 provides data on only 27 women. Looking at the paper itself, it becomes apparent that the recruitment was from a tertiary hospital, and that of the 78 women identified with GDM, only 27 were-screened for T2DM. The 78 is likely a biased sample of the general GDM population, and the 27 are likely a biased sample of the 78. Such a study doesn’t contribute usefully to the authors’ aims. How carefully have other studies been reviewed? 

Response: Apologies to know that the supplementary material was not accessible to the reviewer. The quality assessment of the included studies was carefully performed using the Newcastle Ottawa Scale for cohort studies and the scale proposed by Hoy and colleagues for cross sectional studies. The detailed quality assessment was presented in the supplementary tables 1 & 2 (S1 and S2 tables). We also present the summary of the quality appraisal of the included studies in the supplementary figure 1 (S1_Fig). Of the three studies that have less than 100 women with GDM, one had low risk of bias and the other two had high risk of bias. Our discussion is modified to highlight this more explicitly (line no.264-265) in the revised manuscript.

Comments: Abstract results. ‘The relative risk of T2DM among women with history of GDM from SA and SEA was 13 times higher than’. A relative risk can’t be higher in one group than in another. Only the risk can be higher. 

Response: Thank you for pointing out this. We have now rephrased the sentence.

Comments: Line 68-69. ‘The rate of prediabetes among these women was observed to be between3.9% and 50.9%’. The word ‘rate’ implies a time component, but these look like prevalence data. Please be more precise with language. 

Response: Thank you for pointing out this error. This has been changed (line no.66)

Comments: Line 71. ‘during 6 weeks postpartum to 28 years postpartum’. Replace ‘during’ with ‘from’.

Response: Thank you. This is changed

Comments: Line 229. Relative risk of diabetes appears to have been calculated from the published data, and presumably therefore was unable to adjust for confounders such as age. If this is the case, then the RR is quite possibly very misleading, and should not be presented. 

Response: Yes, this was calculated based on the published data. This is similar to other studies (including the recent publication on the global estimates of T2DM incidence from Vounzoulaki et al, 2020) that have reported the relative risk. We have done so to give a comparative estimate in SA and SEA as opposed to the global estimates presented by earlier studies. However, in view of the reviewer’s concern, we have now moved this Figure to the Supplementary table 3 (S3 Table) and modified the manuscript to highlight the limitations of this observation in the revised version of the manuscript (line 237-239).

Comments: Line 249. ‘indicating that it lies closer to the pooled overall prevalence’. Closer than what? English language use needs to be improved throughout by a native English speaker.

Response: We would like to clarify to the Reviewer that in the Supplementary Figure 2, we performed the sensitivity analysis by excluding one study at a time, which gave the prevalence of prediabetes between 25.0% to 27.7%. This was close to the overall estimate of 25.9% when all the studies were included. This sensitivity analysis gave the reassurance that no specific study had significantly influenced the overall pooled prevalence estimate. We have rephrased the sentence for better clarity in the revised version of the manuscript (line 199-203). The manuscript is modified, and the forest plot is shown in the Supplementary Fig 2 (S2_Fig). The manuscript has been read and improved through by a native English speaker.

Reviewer #2: 

The growing prevalence of type 2 diabetes in SA and SEA regions is a major concern. Gestational Diabetes is well recognised as a major risk factor for future diabetes. However, the lack of data in SA /SEA populations until recently, has been a major limitation to understand the true impact of this problem and to develop effective interventions. Improved understanding of the risks and opportunities to mitigate this can have he benefits for the prevention of T2DM and maternal health. 

This systematic review and meta- analysis addresses, this important issue and provides a fairly good insights on the risk of T2DM in patients with GDM. The work presented here is of good quality but there are a few points that need clarification.

Comments. Abstract: This is good overall. It would be good to provide some data/figures in relation to the last sentence of the results section. 

Response: We thank you for the kind words. In response to the reviewer’s comment, we have now moved the analysis based on follow-up period from Supplementary Figure 3 in the older version to main Figure 3 in the revised version of manuscript. The subgroup analysis based on the follow-up period is included in the results section in the revised manuscript (line 207). 

Comment: Introduction: It is well written and references and places the research in context.

Response: Thank you very much.

Comment: Methods: A lot of detail is provided and the essential, principles of systematic review have been followed. PRISMA guidelines have been adhered to. I would suggest it would be better to describe the combinations of SEARCH terms used to identify the relevant papers. 

Response: We would like to clarify that this information was already provided in the supplementary material 1 (S1 Appendix). It is unfortunate that the reviewers could not access the Supplementary materials. We have raised this issue to the Editor and hope it is accessible in the revised version. 

Comments: It is surprising to see that Chinese data was not considered in this study. Why? What group would Chinese be categorised under by the authors?

Response: We have categorised the countries based on the United Nations Statistics 

 Division, based on which SA & SEA include the following countries, SA – Afghanistan, Bangladesh, Bhutan, India, Maldives, Nepal, Pakistan, Sri Lanka; SEA – Brunei, Cambodia, Indonesia, Thailand, Malaysia, Philippines, Singapore, Vietnam. China was not included as it does not come under this category. 

Comments: Although SA and SEA share many common characteristics, many would regard them as distinct ethnic groups. What is the rationale in combining them? 

Response: We agree that these regions may be regarded as distinct ethnic groups. However, as per the International Diabetes Federation, these two regions have the highest risk of GDM (28%). This was the rationale for us reporting the regional prevalence in SA & SEA. 

Comments: As evident from the data and the meta-analysis, there is considerable heterogeneity between the studies. Moreover, including retrospective, cross sectional and prospective studies together actually weakens the study. Wonder if it would be more prudent to focus on prospective studies that have a non-GDM comparison group. 

Response: Thank you for these comments. Taking the reviewer’s comment on board, we have carried out additional subgroup analysis based on study design and is included in the revised manuscript (line no. 228) and the forest plot is included as supplementary figure 3. We agree that there is significant heterogeneity and there are only three prospective studies which included non-GDM comparison group. We have reflected this (line 301-303) in the discussion section of revised manuscript.

Comments: It is common experience that most GDM patients do not attend follow up. It is therefore difficult to ascertain if all those with GDM were assessed for the duration of the studies and if they had a test to confirm the glycemic status. Lot of attrition can be expected in these situations and can therefore lead to under or over estimation of the true risk. No data is provided regarding this. 

Response: Thank you for this comment, we agree that providing the attrition rates could help the readers understand about the quality of the studies. Hence, we have included the rate of follow-up in table 1, pg no. 8 in the revised manuscript and added a sentence to reflect this in the discussion (281). The follow-up rates are between 31.4% and 95.8%. 

Comments: Also, were there any other tests eg HbA1c used to diagnose pre-diabetes and diabetes? 

Response: Thank you. There was one study (Gupta et al (Ref 38)) that has used HbA1c along with OGTT for the diagnosis of T2DM. We have added this information in line 159 of the revised version of manuscript.

Comments: Given the high prevalence of T2DM in these populations, it is possible that some of the subjects classed as GDM may in fact be T2DM or PRE DIABETES prior to pregnancy. How was this addressed in the studies? 

Response: Undiagnosed diabetes or pre-diabetes is increasingly common in younger women of child-bearing age. We accept the Reviewer’s concern, and this is a challenge. The only way to identify the proportion of this group identified as GDM is by early pregnancy screening. Although this is recommended in many guidelines, this is not carried out routinely in many countries including UK. Most of studies included in this systematic review have only mentioned that the follow-up was conducted in women who underwent screening for GDM during the index pregnancy and only two studies explicitly stated that women without history of T2DM was included in the baseline.

Comments: For the 3 studies that were used for computing Relative Risk, was that based on incident Pre-diabetes or Diabetes? It would be good to see some data on additional risk factors if available eg parity, family history? 

Response: The relative risk was calculated based on the incident T2DM reported in the studies. There is limited evidence on the additional risk factors, as most of the studies have not reported on the incident rates after adjustment of potential risk factors. This point is highlighted as in the discussion section of the revised manuscript (line 310-312).

Comments: A risk score /calculator I mentioned in the manuscript, but I could not find it (apologies if I missed it). 

Response: We apologize for not making it clear. The risk score calculator is freely available as a supplementary material in the open access article by Nishanthi et al. We have made this explicit and highlighted that this is not yet validated in SA and SEA populations, in the discussion section of the revised version of the manuscript (line 288-289). 

Comments: The discussion is well written and balanced. It can be improved by some reference to data in other populations and how these findings compare by providing some figures and data. More so with reference to other ethnic groups with similarly high predisposition to T2DM such as PIMA Indians etc.

Response: Thank you for the kind words. Yes, we have attempted to compare the with other cohorts including the STRONG study. Evidence of the risk of T2DM in woman with history of GDM from the PIMA Indians is lacking. We have reflected this as “However, the uptake rate of postpartum glucose monitoring is sub-optimal even in well developed countries. [24] And also there seem to be limited evidence to compare the prevalence of T2DM post-GDM in different ethnicities including PIMA Indians. A recent study by Napoli et al [25] in Italy, reported that only 34.4% of women from ‘STRONG’ observational study underwent postpartum glucose monitoring. We also observed only three studies had a follow-up rate of more than 70%, although we have showed earlier that it is feasible with a follow up of 95.8%, at least in research setting. [26]”, in the revised manuscript (line 278-281).

---

## [Editor Report · Decision Letter 1]

24 Nov 2022

Prevalence of prediabetes and type 2 diabetes mellitus in south and southeast Asian women with history of gestational diabetes mellitus: systematic review and meta-analysis

PONE-D-22-18537R1

Dear Prof. Saravanan,

We’re pleased to inform you that your manuscript has been judged scientifically suitable for publication and will be formally accepted for publication once it meets all outstanding technical requirements.

Kind regards,

Diane Farrar

Academic Editor

PLOS ONE

Additional Editor Comments:

I have enjoyed reading this informative, very well written and conducted review, I commend you and your colleagues for your hard work

---

## [Editor Report · Acceptance letter]

4 Dec 2022

PONE-D-22-18537R1 

Prevalence of prediabetes and type 2 diabetes mellitus in south and southeast Asian women with history of gestational diabetes mellitus: systematic review and meta-analysis 

Dear Dr. Saravanan:

I'm pleased to inform you that your manuscript has been deemed suitable for publication in PLOS ONE. Congratulations! Your manuscript is now with our production department. 

Kind regards, 

on behalf of

Dr. Diane Farrar 

Academic Editor

PLOS ONE